# Midgut Cell Damage and Oxidative Stress in *Partamona helleri* (Hymenoptera: Apidae) Workers Caused by the Insecticide Lambda-Cyhalothrin

**DOI:** 10.3390/antiox12081510

**Published:** 2023-07-28

**Authors:** João Victor de Oliveira Motta, Lenise Silva Carneiro, Luís Carlos Martínez, Daniel Silva Sena Bastos, Matheus Tudor Candido Santos Resende, Bárbara Monteiro Castro Castro, Mariana Machado Neves, José Cola Zanuncio, José Eduardo Serrão

**Affiliations:** 1Department of General Biology, Institute of Biotechnology Applied to Agriculture, Federal University of Viçosa, Viçosa 36570-900, Brazil; joao.motta@ufv.br (J.V.d.O.M.); lenise.carneiro@ufv.br (L.S.C.); daniel.bastos@ufv.br (D.S.S.B.); matheus.resende@ufv.br (M.T.C.S.R.); mariana.mneves@ufv.br (M.M.N.); 2Faculdad de Ciencias Agricolas, Universidad de Narino, Pasto 522020, Colombia; lcmartinez@udenar.edu.co; 3Department of Entomology, Institute of Biotechnology Applied to Agriculture, Federal University of Viçosa, Viçosa 36570-900, Brazil; barbara.monteiro@ufv.br (B.M.C.C.); zanuncio@ufv.br (J.C.Z.)

**Keywords:** cytotoxicity, histopathology, pollinators

## Abstract

The stingless bee *Partamona helleri* plays a role in pollinating both native and cultivated plants in the Neotropics. However, its populations can be reduced by the pyrethroid insecticide lambda-cyhalothrin. This compound may cross the intestinal barrier and circulate through the hemolymph, affecting various non-target bee organs. The aim of the present study was to assess the extent of cellular damage in the midgut and the resulting oxidative stress caused by lambda-cyhalothrin in *P. helleri* workers. Bees were orally exposed to lambda-cyhalothrin. The lethal concentration at which 50% of the bees died (LC_50_) was 0.043 mg a.i. L^−1^. The *P. helleri* workers were fed this concentration of lambda-cyhalothrin and their midguts were evaluated. The results revealed signs of damage in the midgut epithelium, including pyknotic nuclei, cytoplasm vacuolization, changes in the striated border, and the release of cell fragments, indicating that the midgut was compromised. Furthermore, the ingestion of lambda-cyhalothrin led to an increase in the activity of the detoxification enzyme superoxide dismutase and the levels of the NO_2_/NO_3_ markers, indicating oxidative stress. Conversely, the activities of the catalase and glutathione S-transferase enzymes decreased, supporting the occurrence of oxidative stress. In conclusion, the ingestion of lambda-cyhalothrin by *P. helleri* workers resulted in damage to their midguts and induced oxidative stress.

## 1. Introduction

Animals, especially bees, assume an indispensable role in the pollination process, benefiting over 85% of cultivated angiosperms across the globe [1]. This crucial function makes them irreplaceable for the sustenance of life on our planet [2]. The significance of animals, particularly bees, in facilitating the reproductive success of flowering plants cannot be overstated, as their actions directly contribute to the diversity and abundance of various ecosystems. Due to their pivotal role, it is imperative to safeguard and support these pollinators to ensure the continuity of ecological balance and agricultural productivity.

Among bees, *Apis mellifera* Linnaeus (Hymenoptera: Apidae) holds significant importance in agriculture due to its ability to enhance crop yields by approximately 95% [3]. In Brazil, alongside honey bees, stingless bees (Hymenoptera: Apidae: Meliponini) also serve a crucial role in the ecosystem [4]. These stingless bees exhibit social behavior, display generalist feeding habits, and are commonly found in tropical and subtropical regions [5]. Remarkably, they are responsible for pollinating 90% of the native flora, encompassing economically valuable plants like eucalyptus and citrus [5,6]. Understanding the vital contributions of both *A. mellifera* and Meliponini in plant pollination is essential for sustainable agriculture and preserving biodiversity.

*Partamona helleri* Friese 1900 (Hymenoptera: Apidae: Meliponini), a eusocial stingless bee native to the Neotropical region, exhibits a remarkable characteristic of forming permanent colonies that can house hundreds to thousands of workers. However, this species and other bees are exposed to numerous threats that have contributed to a decline in their populations. Habitat loss, environmental degradation, climate change, lack of proper management [7,8,9], and the use of pesticides [10,11] are the main causes driving bee population decline. Of particular concern is the toxic effect of insecticides on pollinators, including bees. These chemicals can adversely affect bees through direct contact with treated plants or by inhaling airborne toxic particles while flying over contaminated areas [12]. Furthermore, studies have detected insecticide residues in nectar and pollen [13], suggesting that bees may suffer adverse effects through ingestion [14]. Recognizing and addressing these pressing issues is crucial to safeguard the survival and ecological contributions of *P. helleri* and other vital pollinators in the Neotropical region.

Since 2008, Brazil has witnessed a staggering surge in pesticide usage (nearly 200%), occupying the top position in the world consumption ranking [15]. Notably, among the various insecticides employed, pyrethroids constitute a prominent class of neurotoxic compounds, comprising synthetic analogs of the natural pyrethrins derived from the flowers of the Asteraceae *Crisantemum cinerafolis* [16]. Pyrethroids can be categorized into two classes, namely types I and II, with each exhibiting distinct symptomatic effect. However, type II pyrethroids differ from type I pyrethroids due to the presence of an α-cyano group in their chemical structure [14,16]. These insecticides demonstrate remarkable stability when exposed to light and environmental temperature, undergoing degradation through hydrolysis and oxidation [14]. Understanding the characteristics and complexities of pyrethroids is crucial for assessing their impact on the environment and the potential risks to both non-target organisms and human health.

Lambda-cyhalothrin, classified as a type II pyrethroid, possesses lipophilic properties that facilitate its rapid penetration into insect tissues [13,17]. Upon entering the insect body, this compound disrupts the conduction of nerve stimuli and feeding processes, leading to a cascade of detrimental effects. These include the loss of muscle control, resulting in paralysis and ultimately culminating in the death of the insect [17,18]. The potent neurotoxic action of lambda-cyhalothrin has made it an effective tool for pest control in various agricultural and public health applications. However, it is essential to balance its use with environmental considerations and potential impacts on non-target organisms, including beneficial insects like bees and other pollinators. Sustainable pest management strategies that minimize the risks to beneficial insects and ecosystems are critical for safeguarding biodiversity and long-term agricultural productivity.

Upon ingestion, insecticides initiate their interaction with the digestive tract [19], with the midgut being the primary organ responsible for absorption in bees [19]. The bee midgut comprises three distinct cell types: columnar digestive cells, which are crucial for nutrient absorption and digestive enzyme production [20]; endocrine cells, which are responsible for hormone peptide production [21]; and regenerative cells, which are tasked with cell replacement [22]. Additionally, the midgut serves as a site for detoxification following oral exposure to insecticides [23]. Once inside, insecticides cross the midgut epithelial barrier and disperse throughout the hemolymph, inflicting harm on non-target organs in bees, such as the brain, hypopharyngeal glands, fat body, and Malpighian tubules, and significantly impacting insect physiology and behavior [17,24,25,26,27,28]. Studies involving the midgut of honey bees and *P. helleri* have reported structural alterations in the brush border, the release of cell fragments into the gut lumen, and compromised food digestion and absorption after the ingestion of imidacloprid [27] and lambda-cyhalothrin [28].

Insecticides have the potential to trigger the production of hydrogen peroxide (H_2_O_2_) through processes such as the dismutation of the superoxide radical anion (O_2_^−^) by oxidase enzymes or the β-oxidation of fatty acids. This mechanism serves as a crucial cellular defense against oxidative stress, as observed in bees [28,29]. Mitochondria play a pivotal role as the primary source of O_2_^−^, leading to an increase in the levels of the enzyme superoxide dismutase (SOD). SOD converts O_2_^−^ into H_2_O_2_, which is partly eliminated by catalases (CAT) and glutathione S-transferase (GST) [30]. In the context of oxidative stress assessment, scientists often measure reactive nitrogen species levels, particularly nitrites (NO_2_^−^) and nitrates (NO_3_^−^). The conversion of nitrate to nitrite results in the production of nitrous acid (HNO_2_), which can lead to the deamination of DNA bases containing a free -NH_2_ group [30]. The comprehension of the biochemical responses to oxidative stress caused by insecticides is essential for understanding the potential impact of pesticides on bee health.

Given the vital role of bees in pollination and their significant contribution to the environment, it is concerning that the insecticide lambda-cyhalothrin has been detected to contaminate the food consumed by *P. helleri* bees [5], an essential Neotropical pollinator. Additionally, there is a lack of data on the specific impacts of this insecticide on *P. helleri* bees. Therefore, this study aimed to assess whether lambda-cyhalothrin exhibits toxicity against *P. helleri* and whether it causes damage to the midgut and induces oxidative stress within these bees. Understanding the potential risks of lambda-cyhalothrin on vital pollinators like *P. helleri* is crucial for developing informed conservation and regulatory measures to protect these essential species and maintain the delicate ecological balance in the Neotropical region.

## 2. Materials and Methods

### 2.1. Bees

Workers of *P. helleri* (*n* = 210) were collected at the entrance of the nest of three colonies kept at the Central Apiary of the Federal University of Viçosa (UFV), municipality of Viçosa (20°45″ S 42°52″ W) state of Minas Gerais, Brazil. The bees were maintained at 25 °C and 65 ± 5% relative humidity in dark conditions in a laboratory at the Institute of Biotechnology Applied to Agriculture (BIOAGRO/UFV).

### 2.2. Concentration–Mortality

The mean lethal concentration (LC_50_) of the lambda-cyhalothrin (Karate Zeon^®^ CS; 50 g a.i.; Syngenta AG—Basel, Switzerland) was evaluated by diluting this insecticide in distilled water and 50% sucrose solution, obtaining six concentrations: 312 µg, 156 µg, 78 µg, 39 µg, 19 µg, and 10 µg a.i. L^−1^. The bees were kept in 250 mL plastic containers (11 cm in diameter and 8 cm in height) perforated at the top for ventilation and with 1.5 mL perforated feeding tubes [31]. The food, with the insecticide concentrations and 50% sucrose syrup as the control, was available for 24 h, and in the subsequent 48 h, the bees received only 50% sucrose syrup. Three containers were used per concentration, each containing 10 bees from a different colony, totaling 30 workers per concentration for three different colonies.

### 2.3. Histopathology

*Partamona helleri* workers were fed on the estimated LC_50_ of the lambda-cyhalothrin, and 30 alive bees were randomly chosen and evaluated after 24 h (*n* = 10), 48 h (*n* = 10), and 72 h (*n* = 10); those of the control group (*n* = 10) were cryoanesthetized at −5 °C for 3 min and dissected in 125 mM NaCl before their midgut transferred to Zamboni’s fixative solution [32] for 48 h. Then, the midgut samples were dehydrated in a graded ethanol series (70%, 80%, 90%, and 95%) and embedded in glycol methacrylare resin (Leica Biosystem Nussloch GmbH, Wetzlar, Germany) following the manufacturer’s instructions with modifications. Briefly, the midguts were submitted to infiltration in resin/ethanol (1:3 *v*/*v*) for 30 min, transferred to resin/ethanol (1:1 *v*/*v*) for 30 min, following transference to resin/ethanol (3:1 *v*/*v*) for 30 min (all at room temperature). Then, the samples were transferred to pure resin for 16 h at 4 °C, placed in silicone molds filled with resin plus hardener, and polymerized at room temperature for 24 h. Sections with 3 μm thickness were obtained with glass knives in a rotatory microtome Leica RM 2245 (Leica Biosystem Nussloch GmbH, Wetzlar, Germany) stained with hematoxylin (12 min) and eosin (30 s) and subsequently mounted and analyzed using a light microscope Olympus BX60 (Olympus Corporation, Tokyo, Japan). Some midgut slices were submitted to mercury bromophenol blue (100 mL of 2% acetic acid; 0.05 g of bromophenol blue; 1.5 g of mercuric chloride) for 2 h and 15 min and transferred to 0.5% acetic acid for 10 min and washed in water to evidence proteins. Then, the samples were mounted and analyzed using a light microscope Olympus BX60 (Olympus Corporation, Tokyo, Japan).

### 2.4. Analysis of Antioxidant Enzymes

Twenty-four *P. helleri* workers that survived after being exposed to LC_50_ lambda-cyhalothrin were randomly collected after 24 h (*n* = 8), 48 h (*n* = 8), and 72 h (*n* = 8), and those of the control group (*n* = 8) were carefully dissected for midgut removal. Each midgut was individually placed in a microtube containing 1 mL of 0.1 M sodium phosphate buffer (PBS). To ensure proper extraction of the enzymatic contents, the midguts were homogenized using the Tissue Master 125 homogenizer from OMNI. Following homogenization, the samples were centrifuged at 10,000× *g* for 10 min at 4 °C in a 5430R centrifuge (Eppendorf, Enfield, UK). The resulting supernatant was collected and stored at −80 °C for further analysis. Superoxide dismutase activity was determined by using the pyrogallol method, based on the dismutation of superoxide into hydrogen peroxide by this enzyme [33]. The reaction mixture contained 170 μL of potassium phosphate buffer (0,1 M, pH 7.8) and 10 μL of the sample. It was started by adding 20 μL of pyrogallol (50 mM). The final reaction was measured by absorbance at 320 nm. SOD activity was calculated as units per milligram of protein. One unit (U) of SOD was defined as the amount that inhibited the rate of pyrogallol autoxidation by 50%. The activity of the catalase enzyme was determined by the decomposition rate of the H_2_O_2_. Catalase activity was evaluated by incubating the enzyme sample in 1.0 mL of substrate (65 µmol/mL of hydrogen peroxide in 60 mmol/L of sodium potassium phosphate buffer, 1.1 g of Na_2_HPO_4_, and 0.27 g of KH_2_PO_4_ in 100 mL of distilled water, pH 7.4) at 37 °C for three minutes. The reaction was stopped with ammonium molybdate (32.4 mmol/L). A control test without hydrogen peroxide was conducted to exclude interferences. The yellow complex of molybdate and hydrogen peroxide was evaluated at 374 nm against the blank [34]. To calculate CAT activity, a standard curve was built with serial dilutions of hydrogen peroxide. CAT activity was expressed in CAT U/milligrams of protein. The glutathione-S-transferase activity was determined by the rate of formation of the glutathione conjugate with the substrate 1-chloro-2,4-dinitrobenzene [35]. Briefly, 5 μL of 1-chloro-2,4-dinitrochlorobenzene (CDNB) (0.1 M) was added to the buffer containing 5 μL of GSH (0.1 M), 185 μL phosphate saline buffer, and an aliquot (10 μL) of the homogenate supernatant. Upon the addition of CDNB, the alteration was monitored by increasing the absorbance values at 340 nm for 120 s. The molar extinction coefficient used for CDNB was ɛ_340_ = 9.6 mmol/L × cm. GST activity was normalized by total protein content in homogenate and is expressed as U/ milligrams of protein.

### 2.5. Oxidative Stress Biomarkers

Nitric oxide concentrations in 24 random surviving *P. helleri* workers exposed to LC_50_ lambda-cyhalothrin for 24 h (*n* = 8), 48 h (*n* = 8), 72 h (*n* = 8), and the control group (*n* = 8), were determined by using the Greiss method [36]. Briefly, the midguts were homogenized (Tissue Master 125 homogenizer, OMNI) in 1 mL of ice-cold PBS and subsequently centrifuged at 10,000× *g* for 10 min at 4 °C (Hearaeus Fresco 15430R centrifuge, Eppendorf6, Thermo Scientific, Waltham, MA, USA), and 50 μL of the supernatant was transferred to 50 μL of sulphanilamide solution (1% of sulphanilamide diluted in 5%phosphoric acid) into a microplate and incubated for 10 min at room temperature in the dark. Then, 50 μL of 0.1% *N*-(1-naphthyl)-ethylenediamine solution was added to the reaction, and the material was incubated for 10 min and analyzed using a microplate reader (Thermo Scientific Multiskan SkyHigh) with 540 nm wavelengths.

### 2.6. Statistics

The concentration mortality data were submitted to PROBIT analysis to estimate the concentration–mortality density curve using the PROC ROBIT procedure of the SAS program v.9.0 (SAS Institute, Cary, NC, USA). Oxidative stress data were subjected to one-way analysis of variance (one-way ANOVA) with treatment as a fixed effect, and the means were compared post hoc using Tukey’s honestly significant difference (HSD) test at 5% significance level. Residuals were verified for normality and homoscedasticity for all datasets, and no data transformations were necessary.

## 3. Results

### 3.1. Concentration–Mortality

The mortality data for the *P. helleri* workers fitted a concentration–response model (*p* > 0.05), with an estimated LC_50_ of 0.043 mg a.i. L^−1^ (Table 1). The mortality in the control group was <1%.

### 3.2. Histopathology

The midgut epithelium of the *P. helleri* workers was formed by columnar digestive cells with nuclei rich in decondensed chromatin, well-developed apical striated borders, and cytoplasm with a homogeneous aspect (Figure 1). There were also some nests of regenerative cells. The midgut lumen was lined by multiple layers of the peritrophic matrix (Figure 1).

In *P. helleri*, after 24 h, 48 h, and 72 h of exposure to the LC_50_ of lambda-cyhalothrin, the midgut epithelium presented cytoplasm vacuolization (Figure 2a–d), disorganization in the striated border (Figure 2a–d), pyknotic nuclei (Figure 2b), the release of cell fragments towards the gut lumen (Figure 2a,f), and the disruption of the peritrophic matrix (Figure 2b,c,e).

The histochemical test for evidence proteins revealed the midgut epithelium with uniform reaction, including the striated borders and the nuclei in the control group (Figure 3a). The midgut of the *P. helleri* bees exposed to LC_50_ lambda-cyhalothrin for 24 h showed epithelium with unstained vacuoles, striated borders, and nuclei with reactions to the histochemical test (Figure 3b). The histochemical test was stronger at 24 h of exposure to this insecticide than at 48 h and 72 h of exposure (Figure 3c,d).

### 3.3. Antioxidant Enzymes and Oxidative Stress Markers

The superoxide dismutase activity in the midgut of *P. helleri* workers exposed to LC_50_ lambda-cyhalothrin for 72 h was approximately 30 U/mg ptn, higher than those found in bees exposed for 24 h (approximately 10 U/mg ptn) and 48 h (approximately 15 U/mg ptn), which were similar to the control bees (Figure 4A). The catalase activity after 24 h and 48 h of exposure to the insecticide was similar between the treated bees and the control group (approximately 400 U/mg ptn) but decreased after 72 h of exposure to less than 200 U/mg ptn (Figure 4B). The glutathione S transferase activities were lower among all lambda-cyhalothrin-treated bees, approximately 40 U/mg ptn, compared to the control bees, for which the corresponding figure was approximately 60 U/mg ptn (Figure 4C). The values of the NO_2_/NO_3_ biomarkers were similar, approximately 6 μM in the control, and after 24 h and 48 h of exposure to LC_50_, and this decreased to approximately 4 μM after 72 h of exposure (Figure 4D).

## 4. Discussion

The estimated LC_50_ value (0.043 mg a.i. μL^−1^) for *P. helleri* treated with lambda-cyhalothrin is approximately 100-times lower than that reported for the honey bee *A. mellifera* (4.134 mg a.i. μL^−1^) [17], indicating that *P. helleri* is more vulnerable to this pyrethroid. Additionally, the LC_50_ of *P. helleri* treated with fipronil was four times lower than that of *A. mellifera* [28], and the LD_50_ value was higher for the stingless bee *Megachile rotundata* (1130 pg/bee) compared to *A. mellifera* (0.013 µg/bee) [37]. Furthermore, the stingless bee *Tetragonista angustula* Latreille (Hymenoptera: Apidae: Meliponini) demonstrated greater susceptibility to thiamethoxam and imidacloprid compared to *A. mellifera* [38]. On the other hand, the workers of the stingless bees *Scaptotrigona postica* Latreille (Hymenoptera: Apidae: Meliponini) exposed to the active ingredient imidacloprid present a LC_50_ value that is 40 times greater than that of *A. mellifera*, as shown by the authors of [27], indicating that the stingless bee *S. postica* was more resistant than *A. mellifera*. Finally, [25] shows that the estimated concentrations (LC_50_ and LC_10_) of the formulation of the herbicide mixture (Mesotrione + Atrazine-Calaris) are below the recommended concentration for use in the field, presenting a risk to adult workers and colonies of *P. helleri*.

The lambda-cyhalothrin exposure resulted in damage to the midgut cells of *P. helleri*, including disorganization in the striated border, nucleic pyknosis, the destruction of the peritrophic matrix, and the release of cell fragments into the midgut lumen. These effects are similar to those observed in the midgut of *A. mellifera* after exposure to lambda-cyhalothrin [17] and fipronil [28]. Disruptions in the striated border can lead to a decrease in the cell surface area available for nutrient absorption and compromise ion transport [39,40]. The partial destruction of the peritrophic matrix reduces the protection of the midgut epithelium against mechanical damage, pathogens, and toxins [41]. The release of cell fragments may be related to the elimination of dead cells due to the action of the insecticide lambda cyhalothrin [28]. Understanding these cellular mechanisms is essential for comprehending the broader implications of insecticide exposure on insect health.

The presence of heavily condensed chromatin (nuclear pyknosis), apical ridges, and the disorganization of the striated border in digestive epithelial cells treated with the insecticide lambda-cyhalothrin suggests cell necrosis and apoptotic death [42,43], mediated by the action of caspases and the activation of DNA degradation pathways [44,45]. These changes occur due to the phosphorylation of the cytoskeleton by effector caspases [42]. The cell cortex and microvilli of the striated border are maintained by actin filaments, a component of the cytoskeleton [46] which can undergo disorganization mediated by the action of caspases, causing a loss of cell shape and apical protrusions. The nests of regenerative cells found in the epithelium of *P. helleri* workers indicate the production of new digestive cells as a compensatory mechanism for cell death, maintaining midgut homeostasis [46,47]. However, reports have indicated that exposure to fipronil can result in disorganization in the nests of regenerative cells, compromising cell renewal in *P. helleri* [28]. In addition, damage to organelles, such as dilations in the lumen of the rough endoplasmic reticulum, mitochondrial damage, and disorganization of the nuclear envelope, were reported for the same insecticide in *A. mellifera* [17], for fipronil [28], and for a mixture (Mesotrione + Atrazine-Calaris) [25] in *P. helleri* [28], respectively.

The reduction in protein levels induced by lambda-cyhalothrin in the midgut of *P. helleri* is evidenced by the bromophenol blue histochemical test. This reduction was observed by the low reaction of the midgut epithelium over time. The previously stated damage indicators, such as the partial destruction of the striated border and peritrophic matrix [17,27,28], are more evident after 72 h of exposure to lambda cyhalothrin (when the reaction is weaker) compared to 24 and 48 h of exposure.

Oxidative stress induced by lambda-cyhalothrin can lead to various detrimental effects on *P. helleri*, including lipid peroxidation, disruption of the plasma membrane, and damage to DNA and cellular proteins [28,47], which aligns with the findings from the histochemical test and light microscopy. After 72 h of exposure to lambda-cyhalothrin, an increase in superoxide dismutase enzyme activity can be observed, indicating the cell’s attempt at detoxification through the dismutation of superoxide (O_2_^−^) [48]. The higher superoxide dismutase activity, accompanied by a lower concentration of available superoxides, leads to a decrease in nitrite (NO_2_^−^) and nitrate (NO_3_^−^) markers, as superoxides are required for the formation of nitrous acid (HNO_2_) [48].

The catalase enzyme, responsible for the dismutation of hydrogen peroxide (H_2_O_2_) into oxygen (O_2_) and water (H_2_O), exhibits intense activity within the first 48 h after exposure to lambda-cyhalothrin, indicating the initial responses of the cells to eliminate reactive oxygen species and minimize cell damage, thus promoting insect survival [49]. However, despite the elevated catalase activity, the midgut of the bees showed signs of damage, including cytoplasm vacuolization, disorganization in the striated border, nucleic pyknosis, the release of cell fragments towards the gut lumen, and disruptions in the peritrophic matrix. Conversely, the low activity of the glutathione S-transferase enzyme suggests issues in the detoxification process of *P. helleri*. The reduced catalyzation of harmful metabolite conjugation by glutathione S-transferase leads to an accumulation of toxins within the cells, contributing to oxidative stress [50,51]. The reduced activity of GST in bees exposed to the insecticide compromises the phase 2 detoxification process involving glutathione conjugation, potentially leading to the accumulation of toxins and increased oxidative stress. This impairment not only heightens their vulnerability to cellular damage and oxidative stress but also impacts vital physiological processes such as immune function and neurobiology, ultimately affecting the overall health of the bee and the colony [28,52,53,54]. Further research on the GST pathway and its regulation is necessary to enhance our understanding of bee detoxification mechanisms with respect to toxicity mediated by lambda-cyhalothrin.

Despite limited knowledge regarding the precise mechanism through which small insecticide molecules, like the pyrethroid lambda-cyhalothrin, cross the midgut epithelium and access the insect’s hemocoel, it is hypothesized that transcellular diffusion or transport proteins play a role in this process, given their lipophilic nature and low molecular weight [17,19]. These features emerge as plausible explanations for the substantial cellular damage observed in our experimental findings. Further studies in this area are necessary to fully unravel the intricacies of this phenomenon.

## 5. Conclusions

Our results reveal that the insecticide lambda-cyhalothrin is toxic for adult *P. helleri* workers and that the ingestion of lambda-cyhalothrin damages the midgut epithelium and induces oxidative stress and death in *P. helleri* workers. Overall, our results provide important information about the hazards associated with this pesticide’s toxicity to non-target organisms, including the Neotropical stingless bee pollinator, which should be taken in account in ecological risk assessments.

## Figures and Tables

**Figure 1 antioxidants-12-01510-f001:**
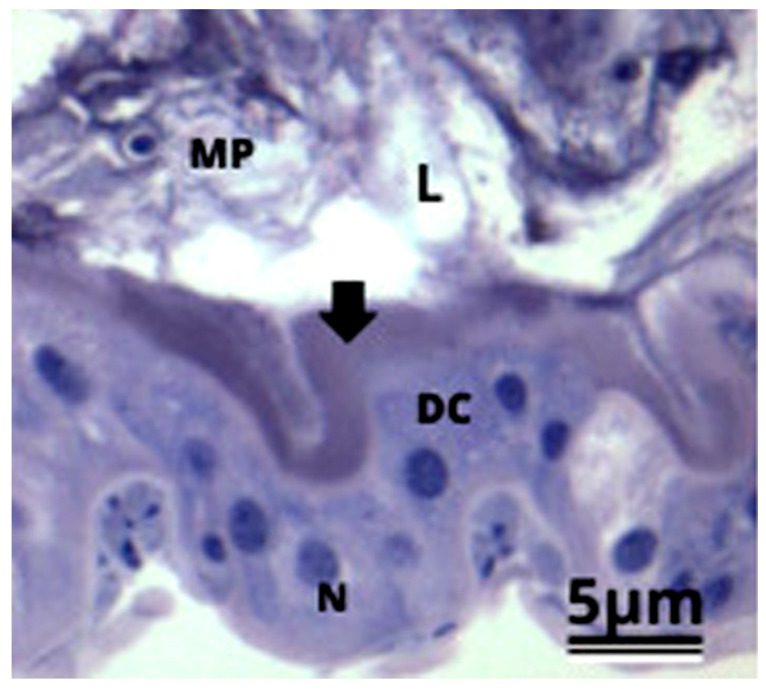
Light micrographs of the midgut epithelium of the stingless *Partamona helleri* (Hymenoptera: Meliponini) bees in the control group, showing columnar digestive cells (DC) with a spherical nucleus (N), striated border (arrow) and the peritrophic matrix (MP) in the midgut lumen (L).

**Figure 2 antioxidants-12-01510-f002:**
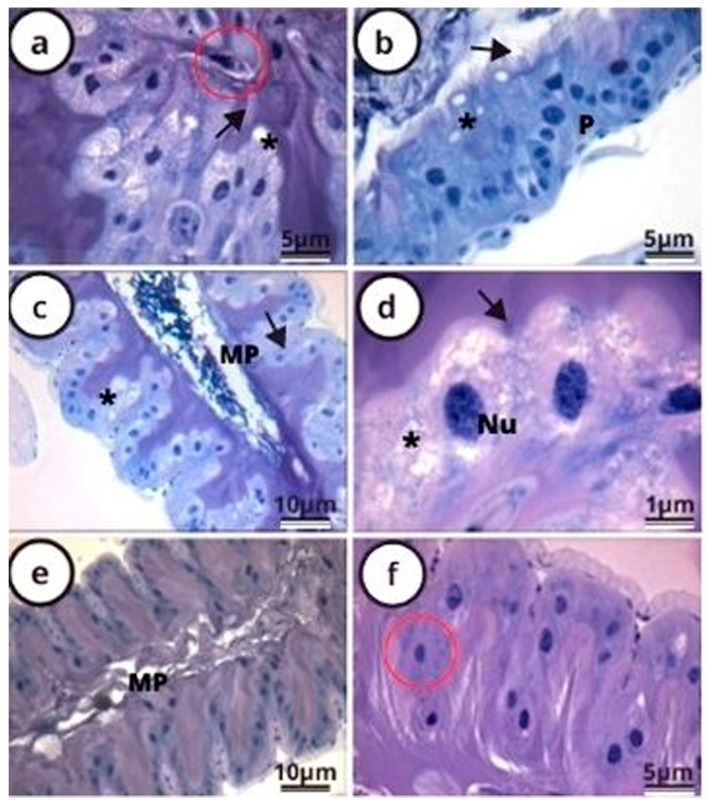
Light micrographs of the midgut of *Partamona helleri* bees (Hymenoptera: Meliponini) exposed to LC_50_ of lambda cyhalothrin for 24 h (**a**,**b**), 48 h (**c**,**d**), and 72 h (**e**,**f**). After exposure, they were stained in H.E. We observed a pyknotic nuclei (P), cytoplasmic vacuolation (*), changes in the striated borders (arrow), cell fragments released into the tissue (red circle), the peritrophic matrix (MP), and nucleolus with condensed chromatin (Nu).

**Figure 3 antioxidants-12-01510-f003:**
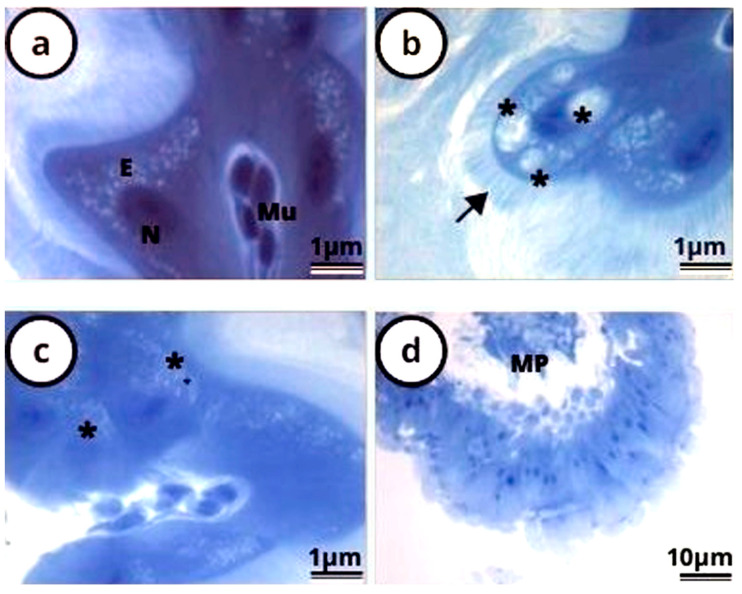
Light micrographs of the midgut of *Partamona helleri* (Hymenoptera: Meliponini) treated in the control group (**a**) and exposed to LC50 of lambda cyhalothrin for 24 h (**b**), 48 h (**c**), and 72 h (**d**) and stained with bromophenol mercury. In the control group, spherical crystals were found in the epithelium (E), well-stained nucleus (N), and musculature of the digestive tract (Mu). In the treatments with 24 h (**b**), 48 h (**c**), and 72 h (**d**), after exposure to lambda cyhalothrin, a light blue striated border (arrow), cytoplasmic vacuolation (*), and the peritrophic matrix (PM) were observed.

**Figure 4 antioxidants-12-01510-f004:**
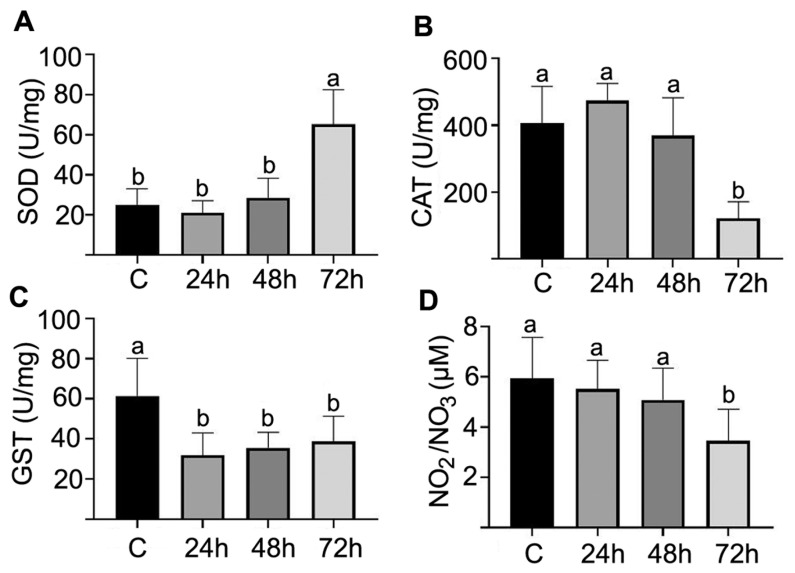
Activity (mean ± standard error-SE) of the enzymes superoxide dismutase (SOD) (**A**), catalase (CAT) (**B**), glutathione S-transferase (GST) (**C**), and nitrite/nitrate (NO_2_/NO_3_) (**D**) content in workers of *Partamona helleri* (Hymenoptera: Meliponini) in the control (**C**) and after exposure to lambda-cyhalothrin after different periods. Different letters on bars indicate differences after Tukey’s test at 5% significance level.

**Table 1 antioxidants-12-01510-t001:** Estimated lethal concentrations (LC) of the lambda-cyhalothrin for orally exposed *Partamona helleri* (Hymenoptera: Meliponini) workers after 72 h, obtained from Probit analysis (d.f. = 5, intercept = 2.7990).

LC	Estimated Concentration (mg a.i. L^−1^)	Confidence Interval 95%	χ^2^ (*p*-Value)
25	0.020	0.013–0.027	4.75 (<0.001)
50	0.043	0.033–0.055	
75	0.092	0.070–0.13	
90	0.181	0.128–0.310	

## Data Availability

Data are contained within the article.

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
