# Peer review of "Midgut Cell Damage and Oxidative Stress in Partamona helleri (Hymenoptera: Apidae) Workers Caused by the Insecticide Lambda-Cyhalothrin"

_antioxidants, 2023, doi:10.3390/antiox12081510_

Round 1
Reviewer 1 Report
My comments:
1) Line 66-67
Statement at line 66-67 has to be equipped with source of such claim. Who found that the presence of the lambda-cyhalothrin has been detected in the food consumed by P. helleri?
2) 2. Materials and Methods
All subtitles have number 2.1
3) line 84
“Each container had 10 bees from each of the three colonies…”, however, ten is not is not divisible by three, therefore, a one bee was extra from each of the three colonies. Each 250 mL plastic container should be filled with 9 or 12 workers to be all three colonies represented equally in one container.
4) Methodology on rearing of worker bees for Histopathology (3 × n=10), Analysis of antioxidant enzymes (4 × n=8), Oxidative stress biomarkers (4 × n=8) was not described. There was described only methodology for assessment of mortality/concentration. Bees from these containers/experiment could not be used for assessments of the other parameters, is not it so?
5) Figure 1 a 3
Peritrophic matrix (PM) and in figures is used “MP”.
6) Line 163
Lumen should be used with adjective “the midgut lumen”.
7) Figure 2
Pyknotic nucleus (P) is not marked in histological pictures.
8) Figure 3
circular muscles (red circle) – the red circle is absent in light micrographs.
9) Discussion
I recommend to consider if following paper would be cited (it is not my paper):
Mayer D.F., Lunden J.D. 1999: Field and laboratory tests of the effects of fipronil on adult female bees of Apis mellifera, Megachile rotundata and Nomia melanderi. Journal of Apicultural Research 38(3–4): 191–197.
This article is dealing with different LD50 of fipronil in honeybee and two solitary bees from different families. Honeybee was paradoxically the most susceptible species in spite of the highest body weight. Your results prove that stingless bee can be much more susceptible to pesticides.
In conclusion:
The manuscript describes results pointed out risks of the pyrethroid insecticide for stingless bee species.
Manuscript is generally well written with minor shortcomings. A minor methodology mistake occured in filling of experimental containers to keep an equal share of workers from all three colonies as explained above.
Let me congratulate to authors for this work deepening knowledge in the field of bee toxicology. I recommend to publish this manuscript.
Author Response
Dear reviewer,
we thank you for your suggestions. All your comments were relevant, and we made the necessary changes. Modified excerpts in the manuscript are in red.
1) Line 66-67
Statement at line 66-67 has to be equipped with source of such claim. Who found that the presence of the lambda-cyhalothrin has been detected in the food consumed by P. helleri?
Answer: The reference has been added “Padilha, AC; Piovesan, B; Morais, MC; Pazini, JB; Zotti, MJ; Botton, M; Grützmacher, A.D. Toxicity of insecticides on neotropical stingless bees Plebeia emerina (Friese) and Tetragonisca fiebrigi (Schwarz) (Hymenoptera: Apidae: Meliponini). Ecotoxicology. 2020; 29; 119–128.”
2) 2. Materials and Methods
All subtitles have number 2.1
Answer: The error has been repaired.
3) line 84
“Each container had 10 bees from each of the three colonies…”, however, ten is not is not divisible by three, therefore, a one bee was extra from each of the three colonies. Each 250 mL plastic container should be filled with 9 or 12 workers to be all three colonies represented equally in one container.
Answer: There were 3 container each one with 10 bees from a specific colony, which was detailed in the text.
4) Methodology on rearing of worker bees for Histopathology (3 × n=10), Analysis of antioxidant enzymes (4 × n=8), Oxidative stress biomarkers (4 × n=8) was not described. There was described only methodology for assessment of mortality/concentration. Bees from these containers/experiment could not be used for assessments of the other parameters, is not it so?
Answer: The bees need to be alive and cannot be the same ones used for different analyses. Surviving bees are randomly chosen. For histopathology were used 30, for antioxidant enzymes 24, and for oxidative markers 24 survived worker bees exposed to the insecticide plus the control. That was detailed in the text.
5) Figure 1 a 3
Peritrophic matrix (PM) and in figures is used “MP”.
Answer: Corrected.
6) Line 163
Lumen should be used with adjective “the midgut lumen”.
Answer: The adjective was added.
7) Figure 2
Pyknotic nucleus (P) is not marked in histological pictures.
Answer: Corrected.
8) Figure 3
Circular muscles (red circle) – the red circle is absent in light micrographs.
Answer: The red circle had been removed from the image, but an error occurred and it remained in the subtitle. The error has been repaired.
9) Discussion
I recommend to consider if following paper would be cited (it is not my paper):
Mayer D.F., Lunden J.D. 1999: Field and laboratory tests of the effects of fipronil on adult female bees of Apis mellifera, Megachile rotundata and Nomia melanderi. Journal of Apicultural Research 38(3–4): 191–197.
Answer: The article is very good, great suggestion. We quoted at the discussion.
We hope that the new version is as expected.
Reviewer 2 Report
The paper shows some deleterious effects of lambda-cyalothrin on a stingless bee. It is generally well written, but could be improved in some details.
Tethal concentration should be allwais indicate as mg a.i. L-1 as at line 128 and not simply as mg as at line 22; see also lines 79-80.
The first paragraph of the introduction could be rather obscure for readers not well acquainted with stingless bees; adding some information about stingless bees, like that they are eusocial insects living in the tropiscs throughout the world, coul help.
Lines 47-54 are a bit confusing; the reader gets the impression that various effects of lambda-cyalothryn are reported, while the studies cited (references 15-19) deal with other pesticides. The following paragraph is much more clear and could be used as a model.
Line 103: did you store the supernatant at -80 °C?
LIne 114: remove the first 4 °C.
Captions of figures 2 and 3 are a bit confusing and should be made more understandable.
Author Response
Dear reviewer,
We are grateful for your suggestions. All your comments were relevant, and we made the necessary changes. Modified excerpts in the manuscript are in red.
Lethal concentration should be always indicating as mg a.i. L-1 as at line 128 and not simply as mg as at line 22; see also lines 79-80.
Answer: Corrected.
The first paragraph of the introduction could be rather obscure for readers not well acquainted with stingless bees; adding some information about stingless bees, like that they are eusocial insects living in the tropics throughout the world, could help.
Answer: We understand the need for more information about stingless bees. Some information has been added, with their respective references, and the writing enhanced.
Lines 47-54 are a bit confusing; the reader gets the impression that various effects of lambda-cyhalothrin are reported, while the studies cited (references 15-19) deal with other pesticides. The following paragraph is much clearer and could be used as a model.
Answer: Thanks for the suggestion. Writing has been improved and correct references have been added.
Line 103: did you store the supernatant at -80 °C?
Answer: Yes, the information was added.
Line 114: remove the first 4 °C.
Answer: Action taken.
Captions of figures 2 and 3 are a bit confusing and should be made more understandable.
Answer: Captions were reworded.
We hope that the new version is as expected.